# Maternal and child nutrition programme of investigation within the 100 Million Brazilian Cohort: study protocol

Thais Rangel Bousquet Carrilho ,[1] Natanael de Jesus Silva ,[2,3] Enny Santos Paixão ,[2,4] Ila Rocha Falcão,[2,5] Rosemeire Leovigildo Fiaccone,[2,6] Laura Cunha Rodrigues,[2,7] Srinivasa Vittal Katikireddi ,[8] Alastair H Leyland,[8] Ruth Dundas,[8] Anna Pearce,[8] Gustavo Velasquez-Melendez,[9] Gilberto Kac,[1] Rita de Cássia Ribeiro Silva ,[2,5] Mauricio L Barreto,[2,10] On behalf of the Nutrition Study Group of the Center for Data and Knowledge Integration for Health (GeNut-CIDACS)

**Correspondence to**
Dr Rita de Cássia Ribeiro Silva;
rcrsilva@ufba.br

## ABSTRACT

**Introduction** There is a limited understanding of the early nutrition and pregnancy determinants of short-term and long-term maternal and child health in ethnically diverse and socioeconomically vulnerable populations within low-income and middle-income countries. This investigation programme aims to: (1) describe maternal weight trajectories throughout the life course; (2) describe child weight, height and body mass index (BMI) trajectories; (3) create and validate models to predict childhood obesity at 5 years of age; (4) estimate the effects of prepregnancy BMI, gestational weight gain (GWG) and maternal weight trajectories on adverse maternal and neonatal outcomes and child growth trajectories; (5) estimate the effects of prepregnancy BMI, GWG, maternal weight and interpregnancy BMI changes on maternal and child outcomes in the subsequent pregnancy; and (6) estimate the effects of maternal food consumption and infant feeding practices on child nutritional status and growth trajectories.

**Methods and analysis** Linked data from four different Brazilian databases will be used: the 100 Million Brazilian Cohort, the Live Births Information System, the Mortality Information System and the Food and Nutrition Surveillance System. To analyse trajectories, latent-growth, superimposition by translation and rotation and broken stick models will be used. To create prediction models for childhood obesity, machine learning techniques will be applied. For the association between the selected exposure and outcomes variables, generalised linear models will be considered. Directed acyclic graphs will be constructed to identify potential confounders for each analysis investigating potential causal relationships.

**Ethics and dissemination** This protocol was approved by the Research Ethics Committees of the authors' institutions. The linkage will be carried out in a secure environment. After the linkage, the data will be de-identified, and pre-authorised researchers will access the data set via a virtual private network connection. Results

## STRENGTHS AND LIMITATIONS OF THIS STUDY

⇒ The large sample size and diversity of populations to be used in this study, including data of minorities such as Indigenous and *quilombolas* (descendants of people who were slaved), the possibility to stratify the analyses according to several intersecting socioeconomic characteristics and the use of longitudinal data are unique aspects of this nutrition research programme.

⇒ The analytical approaches include cutting-edge methods to analyse growth trajectories, such as the superimposition by translation and rotation and the broken stick models, and to create predictive models using machine learning techniques.

⇒ Data from administrative data sets will be analysed, which were not produced for research and, therefore, may be more prone to measurement errors when compared with data primarily collected for research purposes.

⇒ The study is potentially vulnerable to linkage bias since data from several different administrative systems will be linked.

⇒ The coverage of some national information systems is restricted to socioeconomically disadvantaged populations and varies in quality and completeness across regions of Brazil, which will enable estimates for specific groups, but results may not be generalisable.

will be reported in open-access journals and disseminated to policymakers and the broader public.

## INTRODUCTION

Maternal nutritional status before, during and after pregnancy has been associated with the risk of adverse outcomes for the infant and the mother.[1–4] Prepregnancy overweight,

excessive gestational weight gain (GWG) and interpregnancy weight increase are associated with outcomes such as the birth of large for gestational age infants, neonatal mortality, maternal gestational diabetes, hypertensive disorders and mortality, and in the long-term, childhood and maternal obesity.[1–13] Thus, the increase in the prevalence of such predictors among women of reproductive age in low- and middle-income countries (LMICs) may elevate the risk on the referred outcomes.[14] In Brazil, more than 60% of adult women are overweight.[15] Thus, the association between maternal nutritional status and adverse outcomes, especially in the medium and long-term, needs to be further explored.

The association of maternal interpregnancy weight change with outcomes in subsequent pregnancies can also be of great importance. However, few studies have explored this relationship due to the need for long-term follow-up.[16] Studies show that interpregnancy increases in body mass index (BMI) may increase the risk of short-term outcomes, such as gestational diabetes, hypertensive disorders, caesarean delivery and thromboembolism in the subsequent pregnancy.[9 17] It has also been shown that BMI reduction can increase the risk of small for gestational age birth.[17] However, little is known regarding the association of prepregnancy BMI, GWG, postpartum weight retention, BMI change during pregnancies and important long-term adverse outcomes, such as maternal and child mortality and childhood obesity, in developing and socioeconomically vulnerable populations. In addition, maternal weight and child growth patterns and trajectories remain underexplored in these populations, mainly due to the lack of longitudinal nationally representative data. It is crucial to identify those trajectories, their determinants and how they are related.

Furthermore, studies investigating the influence of maternal and infant diet on the nutritional status of children under the age of 5 based on national population data are also scarce and could help shape maternal and child health services and policy interventions. In general, most available studies evaluated the association between maternal and child diet. Bjerregaard et al[18] studied 19 582 mother–offspring pairs in the Danish National Birth Cohort. They observed a significant association between the maternal diet quality during pregnancy and the quality of the offspring's diet at the age of 14. The results from another cohort, conducted among 3422 mothers and children from Portugal, revealed that children whose mothers had worse dietary scores were more likely to follow unhealthier patterns at the age of 4.[19] Thus, maternal and infant diets seem to be related and to affect a child's nutritional status. However, further investigation to tease apart the complex pathways between the mother and infant's diet and child nutritional status is required.

The use of data from national administrative systems has increased throughout the years.[20] The usage of those data for research purposes has recently received attention with improving methods to analyse big data, enhanced secure storage infrastructure and increased computational capacity.[21 22] Using data collected as part of the routine of healthcare services can be helpful to monitor trends in health indicators and inform public health policies.[23] These data are critical in LMICs, where investment in research is generally scarce. Since 1975, several information systems have been created in Brazil. However, little has been done to harness their resources, especially in the maternal and child nutrition field. Big administrative databases can make it possible to study several maternal and child nutritional health problems which have only been investigated in studies with small sample sizes or not explored at all.

Maternal nutritional status, diet and lifestyle, and the family's sociodemographic conditions are all potential determinants of the occurrence of maternal and infant adverse outcomes. However, these relationships were not adequately tested in LMICs. In addition, these are complicated pathways, and causal methods, applied to longitudinal data and through a life course lens, along with large sample sizes, can help in the understanding of those. Thus, the current protocol represents an unprecedented study in an LMIC population, and it differs from currently available studies carried out in small cohorts and with restricted geographical representation. To the best of our knowledge, this is the first initiative to use a population-based linkage database of administrative and health registry data to study maternal and child health and nutritional outcomes in Brazil.

Our objectives in this nutrition research programme are to: (1) describe maternal body weight trajectories during the life course (adolescence, adulthood, pregnancy and postpartum); (2) describe child weight, height and BMI trajectories until the age of five; (3) create and validate models to predict childhood obesity at the age of 5 years; (4) Estimate the effects of prepregnancy BMI, GWG, maternal weight and BMI changes between pregnancies, and maternal weight trajectories on adverse maternal and neonatal outcomes and child growth trajectories; (5) estimate the effects of prepregnancy BMI, GWG and maternal weight and BMI changes on adverse maternal and neonatal outcomes in the subsequent pregnancy; (6) estimate the effects of maternal and infant feeding practices (food intake markers during pregnancy, breast feeding and complementary feeding) on child nutritional status and growth trajectories. We expect that the results obtained from these studies can be translated into actions to improve women's and children's healthcare and public health policy, and in the long-term, in an improvement in the health indicators in these groups.

## METHODS AND ANALYSIS
### Conceptual framework
Our conceptual framework considers maternal nutritional status, diet and lifestyle, and the family's sociodemographic conditions as potential determinants of the occurrence of adverse maternal and child outcomes (figure 1). The complex relationships underlying this

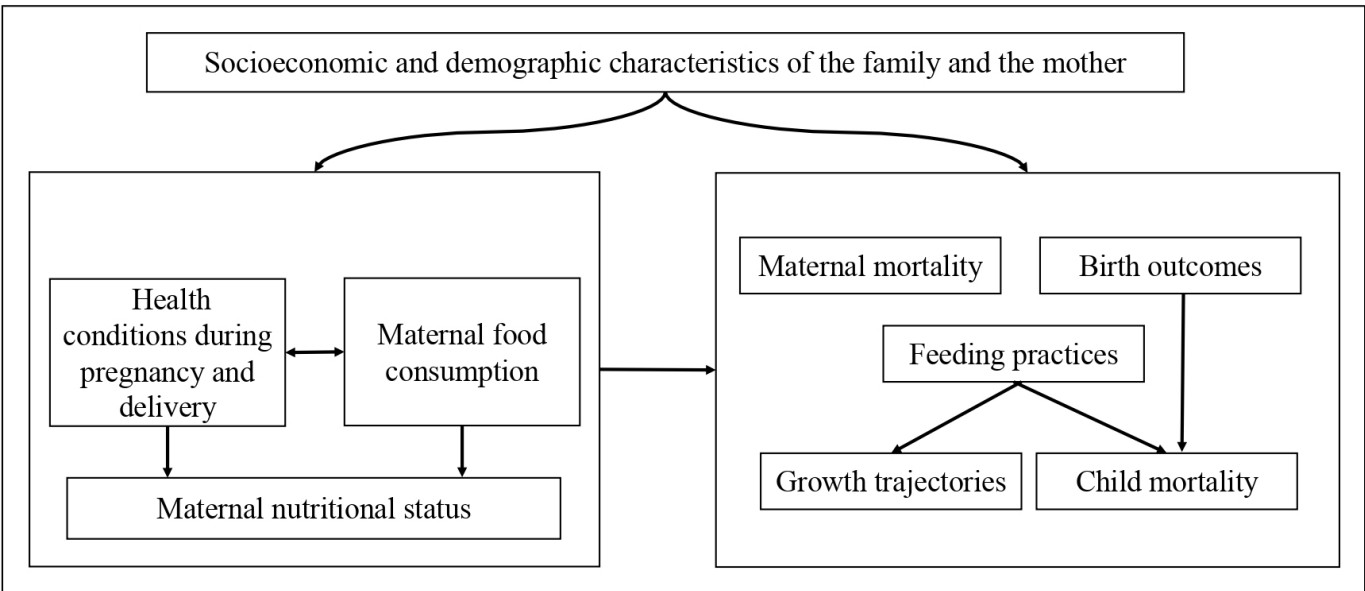

**Figure 1** Theoretical framework guiding the investigation of maternal and child nutrition within the 100 Million Brazilian Cohort.

framework have been poorly investigated in LMICs. Most evidence regarding these associations comes from high-income countries or was conducted with small sample sizes in LMIC populations.[3 11 24]

### Data source and linkage process

This protocol was developed following the Reporting of Studies Conducted Using Observational Routinely Collected Health Data guidelines.[25] This study will use linked data from four different Brazilian databases: the 100 Million Brazilian Cohort baseline, the Live Births Information System (SINASC, from the Portuguese acronym), the Mortality Information System (SIM) and the Food and Nutrition Surveillance System (SISVAN) (table 1). The 100 Million Brazilian Cohort is a dynamic and open cohort created by the *Centro de Integração de Dados e Conhecimentos para Saúde* (CIDACS).[26] The cohort was designed to study the effect of social determinants and social policies on the health of Brazilian people, mainly those who are beneficiaries of social protection programmes. It includes individual data from different national databases from January 2008 to December 2018. The cohort is based on the *Cadastro Único* (CadUnico), a shared registry for more than 20 Brazilian social programmes, containing data from families with a monthly income less than or equal to of three minimum wages (~US$750). The CadUnico database originates from an extensive questionnaire completed at the time of application to federal social programmes and contains detailed demographic, economic and social data at the household and individual levels. The CadUnico database was used to generate the baseline of the 100 Million Brazilian Cohort. More details of this process can be found elsewhere.[27 28] The baseline registers of the cohort comprise 131 697 800 individuals.

The SINASC is the primary national system recording the data available in the 'declaration of live birth', legally required for every live birth in Brazil. The declaration is

filled in immediately after birth by the health worker who assisted the childbirth. In cases where the delivery was not assisted or performed with a registered professional, the declaration is filled in by the Civil Registry Office with the data provided by the declarant. The declaration form includes data on pregnancy and delivery. Details of the system can be found elsewhere.[29] CIDACS has access to identified data of the SINASC between 2001 and 2018, comprising more than 52 million records.

The third database, the SIM, records data from the 'declaration of death', which is a document that must be completed by a physician who attests to the cause of death. The form contains information on the identity of the deceased, their place of residency and death and the cause of death. For infant mortality under 1 year of age, it also provides information on the deceased mother. The details of the death certificates can be found elsewhere.[30] The database available at the CIDACS includes 20 366 604 records between 2001 and 2018.

Finally, the fourth database is the SISVAN, which monitors the Brazilian population's nutritional status since 2008.[31] The system was designed to record individual sociodemographic, anthropometric and food intake data of subjects using public healthcare services in all life-course stages. The main goal of the SISVAN is to inform the evaluation and development of public health nutrition policies. The data set available at CIDACS includes 307 245 508 records (59 724 164 individuals) in all the life cycles from 2008 to 2017. Women of all ages and children aged up to 10 years correspond to 76.4% (235 801 691) and 37.6% (115 510 621) of the records in the SISVAN.

For the work described in this protocol, we will link the 100 Million Cohort baseline data with the data from the SISVAN, SIM and SINASC. The 100 Million Brazilian Cohort will form the population spine for linkage. Two approaches will be used to perform record linkage

Table 1  Variables available in each data source to be used in the maternal and child nutrition research programme

| Database | Data collection process | Available variables | Coverage of the database | Period available |
|---|---|---|---|---|
| 100 Million Brazilian Cohort baseline | The baseline of the 100 Million Brazilian cohort was created using administrative records from individuals ≥16 years of age whose families applied for social assistance via the *Cadastro Único para Programas Sociais*. | Characteristics of the registered individuals (year of application/entry, gender, age at entry year, self-declared race/ethnicity, education, marital status, head of the family), characteristics of the household (municipality of residence, urban/rural area of residence, household building material, water supply, sewage disposal, waste disposal/garbage collection, electricity, number of individuals in the household, number of rooms in the household); 'Bolsa Familia*' information (identification of beneficiary families or not, receipt start date, final receipt date, receipt time in days). | More than 50% of the Brazilian population | 2001–2018 |
| SINASC (Live Births Information System) | The form (declaration of live birth) is completed by a health professional present at the delivery, and the system records information about all live births in Brazil. | Characteristics of the newborn (sex; Apgar score in the 1st and 5th minute, birth weight, presence of abnormalities, congenital anomalies identified at birth using the ICD-10 code), place of birth, characteristics of the mother (name, age, marital status, education, race, place of residence), father's age, characteristics of pregnancy and delivery (number of previous pregnancies of live births, stillbirth or abortion, length of gestation, type of delivery, number of fetuses, number of visits to prenatal care facilities, month of pregnancy in which prenatal care started, which health professionals were present at the delivery). | Approximately 97% of the Brazilian live births | 2001–2018 |
| SIM (Mortality Information System) | The form (declaration of death) is completed by the physician who attests the cause of death. The system records all deaths in Brazil, and it differentiates between fetal and non-fetal deaths. | Characteristics of the dead person (date of death and birth, name, name of the mother and father, sex, race, marital status, occupation and education, address), identification of the place of death (hospital, home, public place and includes the address of the place), characteristics of the mother (name, age, marital status, education, occupation, race, number of births, place of residence, length of gestation, number of previous stillbirths or abortions, type of delivery, number of fetus in the current pregnancy—filled in case of fetal death or infant mortality), birth weight (filled only in case of fetal deaths or infant mortality) and cause of death using ICD-10 code. | SIM coverage has been improving over time, reaching around 95% in some regions†. | 2011–2018 |
| SISVAN (Food and Nutrition Surveillance System) | The forms are completed by primary care workers in the public healthcare system. Anthropometric and food intake markers data of the Brazilian Unified Health System (SUS) users are collected. | Date of birth, age; sex, race/ethnicity, education, participation in social/health programmes, diseases and intercurrences, food intake data (food markers according to each stage of the life cycle) and anthropometric data: weight and height, birth weight for children under 2 years, waist circumference for adults, calf circumference for elderly and prepregnancy weight for pregnant women; | About 30% among under-5 children and 17% among pregnant women | 2008–2017 |

*Bolsa Familia is the conditional cash transfer programme implemented in Brazil since 2004.
†In 2010, the system's coverage was estimated to be 95% in some Brazilian regions.[58]
ICD-10, International Statistical Classification of Diseases and Related Health Problems 10th Revision.

between the four sources: (1) deterministic linkage and (2) non-deterministic linkage based on the similarity index. Deterministic linkage will be performed between the 100 Million Brazilian Cohort baseline and the SISVAN because both data sets contain the Social Information Number (NIS), a unique identifier assigned for each individual. Linkage of the other administrative data sets will occur using a non-deterministic approach. This

method will also link the data from the cohort with the subset of individuals in the SISVAN with no NIS. To do that, CIDACS created a tool (CIDACS-RL) to link individual records based on identifiers, such as name, sex, age or date of birth, mother's name and the municipality of residence.[32 33] The CIDACS-RL applies indexing and searching algorithms implemented in the Apache Lucene solution as the blocking strategy. The indexation strategy allows the CIDACS-RL to search for the most similar records from the indexed data set for each record in the main data set and submit them to the pairwise comparisons step. Candidate linking records are ordered by the similarity scores, and only the comparison pair with the highest score is retained as a potential link. All remaining candidate records are discarded.[32 34] The CIDACS-RL tool was previously validated and presented higher accuracy when compared with other linkage tools.[33]

The final linked data set will be submitted to an extensive data structuring process before the analyses are performed. We will first identify the mother–child pairs in the 100 Million Brazilian Cohort baseline based on the child's family ID number and birth information, such as the mother's name and date of birth. This identification enables the structuration of the data set and the link between the mother and child data. This step is essential because the mother–child pairs are not automatically identified in the linkage.

## Main variables

The main variables for analyses are summarised in table 2. Several maternal and child variables can be created with the linked data sets. These variables will be analysed as outcomes or exposures/confounders for several objectives proposed in this protocol. It is important to mention that, whenever possible, we plan to explore those variables as continuous and categorical, according to the proposed cutoffs. For example, children's BMI will be tested as a continuous score and categorised into underweight, normal, overweight and obesity. Other essential variables to achieve the proposed objectives include socioeconomic and demographic conditions, newborn, mother, pregnancy and delivery characteristics (table 1).

## Analytical approaches

The final linked data set will be submitted to an extensive data-cleaning process before the analyses are performed. In this step, we plan to implement a routine to identify implausible anthropometric values considering cross-sectional and longitudinal data. The approaches proposed by Yang and Hutcheon,[35] based on conditional centiles; Shi et al,[36] based on jackknife residuals; and the method based on linear mixed spline regression proposed by Welch et al[37] and Boone-Heinonen et al[38] will be considered. In addition, modifications of the methods to deal with cross-sectional data[39] and methods to identify populational outliers based on z scores of reference charts will also be considered.[38] After the identification, we plan to perform sensitivity analyses to assess the impact of removing the observations flagged as outliers from the data set.

Finally, because Brazil is a remarkably diverse country, we plan to perform a heterogeneity assessment of the main variables to be used in the studies across the states and regions to ensure we are working with homogeneous data. Two approaches will be considered in this step: (1) the use of multilevel models to calculate the proportion of the variance in the variable of interest that could be attributable to the origin of the data,[39] in this case, the region of residence and (2) the standardised site difference method, as proposed by the WHO in the Multicentre Study.[40] This evaluation of heterogeneity will also capture aspects related to the quality and coverage of the data set, which will be considered before deciding the next steps of the analyses.

We plan to conduct descriptive analyses initially, including the presentation of summary measurements (mean, SD, median, IQR, kurtosis and skewness measurements) for continuous variables and absolute (n) and relative (%) frequencies for categorical ones. A summary of the proposed analytical approaches for each goal is available in table 3.

Because we plan to work with large sample sizes, it will be possible to adopt an intersectionality perspective, exploring how combinations of social characteristics (such as gender and race/ethnicity or socioeconomic position and race/ethnicity) are related to outcomes of interest.[41] We will do that by stratifying the analyses according to several variables, such as those identifying different social contexts. For example, we will be able to stratify the analyses by Brazilian municipality, identifying clusters of richer (closer to the European reality) and poorer (closer to sub-Saharan Africa context) populations and explore how combinations of family size, race/ethnicity, urban/rural context influence the outcomes.

To describe trajectories in maternal weight and child weight, height and BMI, we plan to test three different classes of models: latent class growth, broken stick and Super Imposition by Translation And Rotation (SITAR) models. The novelty of this study is the possibility of stratifying the trajectories according to several sociodemographic and economic variables, such as race/ethnicity and participation in conditional cash-transfer programmes.

We plan to use latent-class growth models to identify trajectory groups that can be later associated with risk factors.[42] Those models allow longitudinal data to inform clusters of individuals following different patterns of change over time, and they account for varying distances between the measurements. The number of trajectory groups can be selected based on Bayesian information criteria.[42]

The *broken stick* models are another recent approach to be used to model growth trajectories. These models are a particular case of the linear mixed model. They summarise irregular individual trajectories, that is, non-equidistant measurements over time, by estimates made

**Table 2** Main variables to be used in the maternal and child nutrition research programme

| Variable | Characteristics |
|---|---|
| **Maternal variables** | |
| Mode of delivery | 1. It is collected as vaginal or caesarean.<br>2. Since 2011, there is also the information if the labour was induced. |
| Maternal pre-pregnancy BMI | ▶ It is calculated based on weight (kg) and height (m) using the formula: BMI weight (kg)/height (m)$^2$.<br>▶ Self-reported prepregnancy weight will be used for its calculation, considering the good agreement observed between self-reported and measured first-trimester weight in a previous study using the SISVAN data.[59]<br>▶ Continuous variable (kg/m$^2$).<br>▶ The WHO cutoffs will be used: underweight (<18.5 kg/m$^2$), normal weight (≥18.5 and <25.0 kg/m$^2$), overweight (≥25.0 and <30.0 kg/m$^2$) and obesity (≥30.0 kg/m$^2$).[60] |
| Maternal weight changes between pregnancies | ▶ It is calculated as the difference in self-reported prepregnancy weights between consecutive pregnancies.<br>▶ Continuous variable (kg).<br>▶ Percentage of weight change.[61] |
| Maternal BMI change between pregnancies | ▶ It is calculated as the difference between BMIs at the beginning of consecutive pregnancies.<br>▶ It will be used as a continuous variable (units of increase/decrease in BMI) or categorised.[62]<br>▶ The change in BMI category (based on the WHO cutoffs) between pregnancies will also be used.[63] |
| Gestational weight gain (GWG) | ▶ Several ways to calculate GWG will be tested in this study.<br>▶ Total GWG is calculated as the difference between the weight measured up to 14 days before delivery and self-reported prepregnancy weight.[64]<br>▶ Cumulative GWG is calculated as the difference between GWG measured in prenatal visits and self-reported prepregnancy weight.[64]<br>▶ Continuous variable (kg).<br>▶ Z scores of the Brazilian GWG charts.[65]<br>▶ GWG per trimester can also be considered. |
| Postpartum weight retention | ▶ It is calculated as the difference between the weight measured at 6 and 12 months postpartum and self-reported pre-pregnancy weight.<br>▶ Continuous (kg) variable.<br>▶ Categorical variable—we plan to test several cutoffs for its classification. |
| Maternal weight trajectories | ▶ This outcome will be created in objective 1, and the groups of trajectories identified will be used in the following studies. |
| Food intake markers | ▶ The following markers are available: consumption of beans; fresh fruits; vegetables; hamburgers or processed, mixed meat products; sugar-sweetened beverages; instant noodles, packaged snacks, or crackers; stuffed biscuits/cookies or candies.<br>▶ The response options are 'yes/no', and the time frame always refers to the intake on the day before the visit. These markers are collected before, during and after pregnancy. |
| Maternal mortality | ▶ Maternal death will be defined as the death of women during pregnancy or up to 42 days after delivery, due to any cause related to or aggravated by the pregnancy, but not due to accidental or incidental causes.[66] |
| **Child variables** | |
| Gestational age at birth | ▶ This information is available in weeks since 2011. Before, it was only available categorised as <22 weeks, 22–27 weeks; 28–31 weeks; 32–36 weeks; 37–41 weeks; and 42 weeks or more.<br>▶ It can be used in weeks or classified as preterm (<37 weeks)/term birth (≥37 weeks).[66]<br>▶ Preterm birth can also be divided into extremely preterm (<28 weeks), very preterm (28–<32 weeks) and moderate or late preterm (32 to <37 completed weeks).[66] |
| Birth weight | ▶ Continuous, in grams.<br>▶ Classified as large (>P90) and small (<P10) for gestational age, according to Intergrowth-21st sex and gestational age-specific charts.[67]<br>▶ Classified as low birth weight (LBW, <2500 g) and macrosomia (>4000 g).[68]<br>▶ LBW can also be classified into very LBW (<1500 g) and extremely LBW (<1000 g).[66] |

Continued

**Table 2** Continued

| Variable | Characteristics |
|---|---|
| BMI up to 5 years old | ► It is calculated based on measured weight (kg) and height (m).<br>► Continuous variable (kg/m$^2$).<br>► BMI-for-age z-scores for sex according to the WHO charts[68] will be used.<br>► It can also be used categorised as proposed by the Brazilian Ministry of Health[69]: Underweight (BMI-for-age <−2 z score); normal (BMI-for-age ≥−2 and ≤2 z score); overweight (BMI-for-age >2 and ≤3 z score); obesity (BMI-for-age >3 z score). |
| Rapid infant growth | ► It is determined based on weight-for-age z scores according to the WHO charts.[68]<br>► It will be classified when the z scores in weight-for-age between two different ages up to 5 years old is >0.67.[70] |
| Growth trajectories | ► This outcome will be created in objective 2, and the groups of trajectories identified will be used in the following studies. |
| Food intake markers | The following markers are collected:<br>► For children <6 months old: consumption of breast milk; porridge; water/tea; cow milk; infant formula; fruit juice; fruits; salt food; other food/beverages.<br>► For children 6–23 months old: consumption of breast milk; whole fruit in pieces or mashed (and the daily frequency); salt food (and the daily frequency and how the food was offered—in pieces/kneaded/passed through a sieve/blended/only the broth); milk other than breast milk; milk porridge; yoghurt; vegetables; orange-coloured vegetables (carrots, pumpkin, papaya) or fruit or dark green leaves (spinach, kale); leaf greenery (lettuce, cabbage); meat (beef, chicken, pork, fish) or eggs; liver; beans; rice, potatoes, yams, cassava, cassava flour or pasta; hamburger and/or sausages; sugar-sweetened beverages; instant noodles, packaged snacks or crackers; stuffed biscuits/cookies or candies.<br>► For children >24 months old: consumption of beans; fresh fruits; vegetables; hamburgers or processed, mixed meat products; sugar-sweetened beverages; instant noodles, packaged snacks or crackers; stuffed biscuits/cookies or candies.<br>► The response options are 'yes/no', and the time frame always refers to the intake on the day before the visit. |
| Child mortality | ► Child mortality refers to the death of children under 5 years old.[66]<br>► It is also possible to work with perinatal, neonatal and postneonatal mortality.<br>► Perinatal mortality includes deaths in the first week of life and fetal deaths (stillbirths).[66]<br>► Neonatal mortality refers to deaths that occurred up to 28 days after birth. It can be divided into two components: (a) Early neonatal mortality, which includes the deaths among live births in the first 6 days of life; (b) late neonatal mortality includes deaths among live births between 7 and 27 days of life.[66]<br>► Post neonatal mortality refers to the deaths among live births between 28 and 364 completed days of life.[66] |

BMI, body mass index; SISVAN, Food and Nutrition Surveillance System.

at a prespecified time interval.[43] The models were initially developed to facilitate the statistical analysis and the testing of critical ages in the onset of childhood obesity, but several other applications are possible.[43 44]

Finally, we plan to test the SITAR models. These models describe each trajectory in three biologically interpretable parameters: size, time and velocity. By using them, we may estimate each child's variation from the curve expressed by child-specific random effects estimates for absolute amount, timing and acceleration.[45] These three child-specific estimates can be used as exposure variables to explore associations with adverse outcomes.[46] It is important to mention that those three techniques are used for different goals, which should be considered when choosing the final model. To select the final model to create the trajectories that will be used in the subsequent studies, we will consider the group's existing knowledge and experts' opinions.

To construct predictive models for childhood obesity at the age of 5, to identify factors that could be used to target interventions, we plan to consider artificial intelligence models, such as those based on Artificial Neural Networks and Adaptive Fuzzy Systems.[47 48] The risk and protective factors to be considered in these models are the characteristics of the child, characteristics of the mother and family/environment factors. The following steps will be executed to create and validate the models: (1) data aggregation and preprocessing (filtering and eliminating inconsistencies); (2) data separation in sets of training, validation and testing; (3) training of the models; and (4) results, validation and testing of the models. The performance of the models will be evaluated through the measures of sensitivity, specificity, positive and negative predictive values, overall accuracy and the area under the receiver operating characteristic curves. Experts will also validate the results based on documents

**Table 3** Summary of objectives, outcomes and proposed analytic approaches

| Objectives | Outcomes/main variables | Analytical approaches |
|---|---|---|
| Describe maternal body weight trajectories during the life course (adolescence, adulthood, pregnancy and postpartum). | Maternal weight, age and gestational age/number of postpartum days in each visit. | ► Latent-class growth models. <br> ► Broken stick models. <br> ► SITAR models. |
| Describe child weight, height., and BMI trajectories until the age of 5. | Children's weight, height, BMI (crude and z scores), age in each measurement. | ► Latent-class growth models. <br> ► Broken stick models. <br> ► SITAR models. |
| Create and validate models to predict childhood obesity at the age of 5. | Predictors: A set of predictors will be considered for the models—characteristics of the child, characteristics of the mother and family/environment factors. <br> Outcomes: Childhood obesity at the age of 5 and rapid infant growth at different ages (12, 18, 24, 36, 48 and 59 months). | Machine learning models using: <br> ► Artificial Neural Networks. <br> ► Adaptive Fuzzy Systems. |
| Estimate the effect of prepregnancy BMI, GWG, maternal weight, BMI changes between pregnancies and maternal weight trajectories on adverse maternal and neonatal outcomes and child growth trajectories. | Exposures: Prepregnancy BMI, GWG, weight, and BMI changes. <br> Outcomes: LGA, SGA, LBW, macrosomia, preterm birth, mode of delivery, child BMI and growth indicators, child growth trajectories defined in goal 2, rapid infant growth, child mortality and maternal mortality. | ► Generalised linear models (Logistic and Poisson regression for categorical and linear and quantile regression for continuous outcomes). <br> ► Linear mixed-effects models for longitudinal outcomes. <br> ► Confounding will be considered using directed acyclic graphs. |
| Estimate the effect of prepregnancy BMI, GWG and maternal weight and BMI changes on adverse maternal and neonatal outcomes in the subsequent pregnancy. | Exposures: Prepregnancy BMI, weight and BMI changes. <br> Outcomes: LGA, SGA, LBW, macrosomia, preterm birth, mode of delivery, children's BMI and growth indicators, growth trajectories defined in goal 2, rapid infant growth, neonatal and child mortality and maternal mortality in the consecutive pregnancy. | ► Generalised linear models (Logistic and Poisson regression for categorical and linear regression for continuous outcomes). <br> ► Linear mixed-effects models for longitudinal outcomes. <br> ► Confounding will be considered using directed acyclic graphs. |
| Estimate the effect of maternal and infant feeding practices on child nutritional status and growth trajectories. | Exposures: Food intake markers during pregnancy, breast feeding, and complementary feeding. <br> Outcomes: Child BMI and growth indicators, growth trajectories defined in goal 2 and rapid infant growth. | ► Generalised linear models (Logistic and Poisson regression for categorical and linear regression for continuous outcomes) <br> Linear mixed-effects models for longitudinal outcomes. <br> ► Confounding will be considered using directed acyclic graphs. |

BMI, body mass index; GWG, gestational weight gain; LBW, low birth weight; LGA, large for gestational age; SGA, small for gestational age; SITAR, superimposition by translation and rotation.

and reference frameworks for child obesity assessment. We will also assess the performance of the algorithm for different population subgroups, particularly from an equity perspective, to identify how it works among more socioeconomically disadvantaged and minority ethnic groups.

To estimate the effects of prepregnancy BMI, GWG, maternal weight and BMI changes between pregnancies, and maternal weight trajectories on the selected outcomes, we plan to use generalised linear models (GLMs). For categorical outcomes, logistic and Poisson regressions with robust variance will be considered. For continuous outcomes, linear regression will be used. We also plan to explore quantile regression for continuous outcomes. If the exposure is a longitudinal variable, we plan to account for the intra-subject variability by running linear mixed-effects models to extract the best linear unbiased prediction. This longitudinal prediction of the exposures can then be incorporated into a GLM model.[49] Whenever the outcome is longitudinal, linear mixed-effects models will be used. We will consider the distribution of the variables and perform transformations whenever necessary. In some situations, we may also consider modelling the data with restricted cubic splines[50] or fractional polynomials[51 52] to account for the non-linearity of the relationships between the given variables. After adjusting the

models, we plan to run the proper diagnostics to evaluate the goodness of fit.

To define the set of confounders to be used in the analyses investigating potential causal relationships, we plan to construct directed acyclic graphs (DAGs) using the programme dagitty (http://www.dagitty.net/). The use of DAGs is becoming quite common in the nutrition field due to their ability to portray the assumptions about the relationships between the variables in the causal structures and establishing the minimal set of variables sufficient for adjusting that blocks all backdoor paths and does not open any closed pathways.[53] The DAGs are particularly useful to identify the minimal set of confounders for adjustment in the regression models. In the cases where we plan to have the analyses stratified (according to socioeconomic conditions, eg), a DAG will be constructed for each subgroup. The consideration of potential confounders will be informed by the existing literature and substantive knowledge of the investigators. We also plan to apply sensitivity analyses to consider the impact of unmeasured confounding.[54]

In the studies aiming to test associations between maternal and child characteristics, it is crucial to deal with missing data in the variables included in the models as exposure and confounders. Hence, we plan to consider the mechanism underlying the occurrence of missing data (whether missingness occurred completely at random, at random or not at random), the pattern of missingness (monotonic or non-monotonic), and the most appropriate method to deal with each of those. Techniques such as multiple imputation, inverse probability weighting and full-information maximum likelihood will be considered.[55 56] We also plan to follow the recently published 'Framework for the treatment and reporting of missing data in observational studies' to deal with missing data in a study-based approach.[57] All the proposed analyses will be performed in R, Stata or Python. The latter is important for the development of the aforementioned artificial intelligence models.

The work described in this protocol was initiated in 2023. The data linkage of several of the databases was already conducted, with a success rate of over 75% in the linkage, but others are still to be done in the following months. The complete timeline for the development of this protocol is 5 years.

## Patient and public involvement

Patients and the public were not involved in any step of this study.

## Ethics and dissemination

The objectives described in this protocol were approved by the Research Ethics Committees from the Gonçalo Moniz Institute/Oswaldo Cruz Foundation (CAAE: 56003716.0.0000.0040), the Institute of Collective Health/Federal University of Bahia (CAAE: 41695415.0.0000.5030), the Federal University of Rio de Janeiro (CAAE: 85914318.2.0000.5275;

18447919.3.0000.5264) and the Federal University of Minas Gerais (CAAE: 37534620.3.0000.5149). To ensure data privacy and confidentiality, all data linkage procedures will be conducted in a physically and virtually secure environment and follow strict internal information security measures.[26] Access to linked and de-identified data sets will be granted to authorised researchers only. The analysis will be performed in the CIDACS data analysis environment, a safe virtual infrastructure that provides remote data access and analysis tools. The final linked and de-identified data sets will receive a Digital Object Identifier, and the complete specification of how the data set was created will be made available on request. Manuscripts from studies of this protocol will be submitted to open-access journals. In addition, we will promote stakeholder events and produce policy briefs to disseminate the results to policymakers and the broader public.

**Author affiliations**

[1]Nutritional Epidemiology Observatory, Josué de Castro Institute of Nutrition, Federal University of Rio de Janeiro, Rio de Janeiro, RJ, Brazil
[2]Centre for Data and Knowledge Integration for Health, Gonçalo Moniz Institute, Oswaldo Cruz Foundation, Salvador, BA, Brazil
[3]Barcelona Institute for Global Health, Hospital Clínic, University of Barcelona, Barcelona, Catalunya, Spain
[4]Epidemiology and Population Health, London School of Hygiene & Tropical Medicine, London, London, UK
[5]School of Nutrition, Federal University of Bahia, Salvador, BA, Brazil
[6]Institute of Mathematics and Statistics, Federal University of Bahia, Salvador, BA, Brazil
[7]Department of Infectious Disease Epidemiology, London School of Hygiene & Tropical Medicine, London, London, UK
[8]MRC/CSO Social & Public Health Sciences Unit, University of Glasgow, Glasgow, Scotland, UK
[9]Department of Maternal and Child Nursing and Public Health, Nursing School, Federal University of Minas Gerais, Belo Horizonte, MG, Brazil
[10]Institute of Collective Health, Federal University of Bahia, Salvador, BA, Brazil

**Collaborators** Nutrition Study Group of the Center for Data and Knowledge Integration for Health (GeNut-CIDACS): The following are members of the GeNut-CIDACS – Nutrition Study Group of the Center for Data and Knowledge Integration for Health: Alastair H Leyland, Aline S Rocha, Anna Pearce, André A Mendes, Andrêa J F Ferreira, Audêncio Victor, Camila S S Teixeira, Carolina S Vieira, Craig Anderson, Dayana R Farias, Davide Rasella, Eduardo A F Nilson, Elizabete J Pinto, Enny S Paixão, Flávia J O Alves, Giesy R Souza, Gilberto Kac, Gustavo Velasquez-Melendez, Ila R Falcão, Jéssica Pedroso, Juliana F Mello e Silva, Laura C Rodrigues, Ligia Kerr, Marcia F Almeida, Maria Y T Ichihara, Mariana S Felisbino-Mendes, Maurício L Barreto, Naiá Ortelan, Miriam H Tsunemi, Natanael J Silva, Nathalia S Guimarães, Rafaela S Andrade, Rita C O Carvalho-Sauer, Rita C Ribeiro-Silva, Rosemeire L Fiaccone, Ruth Dundas, Sara A Silva, Srinivasa V Katikireddi, Thais R B Carrilho, Walmir M Caminhas.

**Contributors** TRBC and NdJS contributed equally to this paper and share the first authorship. GK, RdCRS and MLB share the senior authorship. TRBC, NdJS and GK wrote the manuscript with input from ESP, IRF, RLF, LCR, SVK, AHL, RD, AP, GV-M, RdCRS and MLB. GK, GV-M and RdCRS are the main responsible for the design of the research. RdCRS is the coordinator of the GeNut-CIDACS group. MLB is the coordinator of the CIDACS and responsible for granting access to the data sets. The authors from the GeNut-CIDACS group contributed to the revision of the manuscript. TRBC, NdJS and GK had primary responsibility for the final content. All authors read and approved the final manuscript.

**Funding** This study is funded by the Brazilian Ministry of Health (TED 64/2022, process 25000.148278/2022–10). CIDACS receives core support from the Wellcome Trust (202912/Z/16/Z), the Brazilian Health Surveillance Secretariat, the Brazilian Ministry of Health, the State of Bahia, the Research Support Foundation

of the State of Bahia (FAPESB), the Research and Project Funding Agency (FINEP) and the Secretariat of Science and Technology of the State of Bahia (SECTI). SVK acknowledges funding from an NRS Senior Clinical Fellowship (SCAF/15/02), the Medical Research Council (MC_UU_00022/2) and the Scottish Government Chief Scientist Office (SPHSU17). NdJS acknowledges support from the Spanish Ministry of Science and Innovation and State Research Agency through the 'Centro de Excelencia Severo Ochoa 2019–2023' Programme (CEX2018-000806-S) and support from the Generalitat de Catalunya through the CERCA Programme.

**Competing interests** None declared.

**Patient and public involvement** Patients and/or the public were not involved in the design, or conduct, or reporting, or dissemination plans of this research.

**Patient consent for publication** Not applicable.

**Provenance and peer review** Not commissioned; externally peer reviewed.

**ORCID iDs**
Thais Rangel Bousquet Carrilho http://orcid.org/0000-0001-5928-6136
Natanael de Jesus Silva http://orcid.org/0000-0003-3002-1032
Enny Santos Paixão http://orcid.org/0000-0002-4797-908X
Srinivasa Vittal Katikireddi http://orcid.org/0000-0001-6593-9092
Rita de Cássia Ribeiro Silva http://orcid.org/0000-0002-8387-9254

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
