## [Reviewer comments · BMJ Open]

ARTICLE DETAILS

TITLE (PROVISIONAL)	Maternal and child nutrition program of investigation within the 100 million Brazilian Cohort: study protocol
AUTHORS	Carrilho, Thais; Silva, Natanael; Paixão, Enny; Falcão, Ila; Fiaccone, Rosemeire; Rodrigues, Laura; Katikireddi, Srinivasa; Leyland, Alastair; Dundas, Ruth; Pearce, Anna; Velasquez-Melendez, Gustavo; Kac, Gilberto; Silva, Rita de Cássia; Barreto, Mauricio; Centre for Data and Knowledge Integration for Health, Nutrition Study Group (GeNut-CIDACS)

VERSION 1 – REVIEW

REVIEWER	Luo, Zhong-Cheng Lunenfeld-Tanenbaum Research Institute, Obstetrics and Gynecology, Mount Sinai Hospital
REVIEW RETURNED	05-Jun-2023

GENERAL COMMENTS	The authors described a large population-based birth cohort study protocol. In general, it is well-written. About the data linkage (p8-10), may consider some comments on the following questions: what are the expected successful data linkage rates between different data sources? Any plan for validating the validity of the data linkage in a subset of the linked data? Could unlinked subjects have worse outcomes? How missing linkages may affect the comparisons? About the analysis approach - machine learning prediction models (page 14), may clarify which statistical analysis tools and packages will be used.
--

REVIEWER	Bukania, Zipporah Kenya Medical Research Institute
REVIEW RETURNED	06-Jun-2023

GENERAL COMMENTS	A detailed protocol, it's not clear if the country has an identifier to link all the different datasets as proposed. What will be used as the link variable? The methodology should be clear that anthropometry in children under 5 years of age is measured in Z-scores. this is mentioned in the supplementary text but should be included in the methodology section too. The aims in the abstract can be summarised into one main aim
---

VERSION 1 – AUTHOR RESPONSE

Reviewer #1

Dr. Zhong-Cheng Luo, Lunenfeld-Tanenbaum Research Institute, Shanghai Jiaotong University School of Medicine Xinhua Hospital

The authors described a large population-based birth cohort study protocol. In general, it is well-written.

We thank the reviewer for the comments.

About the data linkage (p8-10), may consider some comments on the following questions: what are the expected successful data linkage rates between different data sources? Any plan for validating the validity of the data linkage in a subset of the linked data? Could unlinked subjects have worse outcomes? How missing linkages may affect the comparisons?

Most data linkage between the 100 million Brazilian Cohort and the SISVAN dataset will occur through a deterministic linkage using the Social Information Number (NIS). Initial linkage procedures between these two databases revealed success rates of over 75%.

In addition, the CIDACS-RL tool, which is being used to perform the probabilistic linkage, was previously validated and presented higher accuracy, scalability, and shorter execution time when compared to other linkage tools” (Barbosa et al., 2020 – reference #33 in the manuscript). We do not anticipate any bias in the linkage process that could affect comparisons.

To accommodate some of the reviewer’s concerns, we added the following information to the text:

Page 10:

The CIDACS-RL applies indexing and searching algorithms implemented in the Apache Lucene solution as the blocking strategy. The indexation strategy allows the CIDACS-RL to search for the most similar records from the indexed dataset for each record in the main dataset and submit them to the pairwise comparisons step. Candidate linking records are ordered by the similarity scores, and only the comparison pair with the highest score is retained as a potential link. All remaining candidate records are discarded (32, 34). The CIDACS-RL tool was previously validated and presented higher accuracy when compared to other linkage tools (33).

Page 16:

The work described in this protocol was initiated in 2023. The linkage of several of the databases was already conducted, with a success rate of over 75% in the linkage, but others are still to be done in the following months. The complete timeline for the development of this protocol is five years.

About the analysis approach - machine learning prediction models (page 14), may clarify which statistical analysis tools and packages will be used.

We complemented this information at the end of the ‘Methods and Analysis’ section (page 16).

New text (page 16):

We also plan to follow the recently published ‘Framework for the treatment and reporting of missing data in observational studies’ (TARMOS framework) to deal with missing data in a study-based approach (57). All the proposed analyses will be performed in R, Stata, or Python. The latter is important for the development of the aforementioned artificial intelligence models.

Reviewer #2

Dr. Zipporah Bukania, Kenya Medical Research Institute

A detailed protocol,

it's not clear if the country has an identifier to link all the different datasets as proposed. What will be used as the link variable?

We thank the reviewer for the comments. In the 'Methods and Analysis' section, in 'Data source and linkage process', we mentioned that a deterministic linkage will be performed between the 100 Million Brazilian Cohort baseline and the SISVAN using the Social Information Number (NIS), which is a unique identifier assigned for each individual in those databases. For the other selected administrative datasets, a non-deterministic approach will be considered. In this case, the CIDACS-RL tool, which links individual records based on identifiers such as name, sex, age or date of birth, mother's name, and the municipality of residence, will be used.

The methodology should be clear that anthropometry in children under 5 years of age is measured in Z-scores. this is mentioned in the supplementary text but should be included in the methodology section too.

It seems there is a misunderstanding regarding the reviewer's comment, as we did not present supplementary information for this protocol. The variables available in each dataset are described in detail in Table 1. In Table 2, we described the possibilities to work with the key variables, including calculating z scores for maternal and children's anthropometric data. This is possible due to the key variables described in Table 1. We opted not to mention those details throughout the text to avoid repeating information available on the tables.

The aims in the abstract can be summarised into one main aim.

We thank the reviewer for the suggestion. Since this is a comprehensive protocol with distinct aims regarding the use of maternal and children's data, we opted to keep those aims separate for clarity to the readers.